# Quality and variation of care for chronic kidney disease in Swiss general practice: A retrospective database study

**Levy Jäger** *, **Thomas Rosemann**, **Jakob Martin Burgstaller**, **Oliver Senn**, **Stefan Markun**

Institute of Primary Care, University of Zurich and University Hospital Zurich, Zurich, Switzerland

* levy.jaeger@usz.ch

## Abstract

### Background

Chronic kidney disease (CKD) is a common condition in general practice. Data about quality and physician-level variation of CKD care provided by general practitioners is scarce. In this study, we evaluated determinants and variation of achievement of 14 quality indicators for CKD care using electronic medical records data from Swiss general practice during 2013–2019.

### Methods

We defined two patient cohorts from 483 general practitioners, one to address renal function assessment in patients with predisposing conditions ($n$ = 47,201, median age 68 years, 48.7% female) and one to address care of patients with laboratory-confirmed CKD ($n$ = 14,654, median age 80 years, 57.5% female). We investigated quality indicator achievement with mixed-effect logistic regression and expressed physician-level variation as intra-class correlation coefficients (ICCs) and range odds ratios (rORs).

### Results

We observed the highest quality indicator achievement rate for withholding non-steroidal anti-inflammatory drug prescription in patients with CKD staged G2–3b within 12 months of follow-up (82.6%), the lowest for albuminuria assessment within 18 months of follow-up (18.1%). Highest physician-level variation was found for renal function assessment during 18 months of follow-up in patients with predisposing conditions (diabetes: ICC 0.31, rOR 26.5; cardiovascular disease: ICC 0.28, rOR 17.4; hypertension: ICC 0.24, rOR 17.2).

### Conclusion

This study suggests potentially unwarranted variation in general practice concerning RF assessment in patients affected by conditions predisposing for CKD. We further identified potential gaps in quality of CKD monitoring as well as lower quality of CKD care for female patients and patients not affected by comorbidities.

**Data Availability Statement:** Data cannot be shared publicly because of possible identification of patients by individuals or organisations with access to overlapping data sets (such as by linkage with

health insurance claims data). Data are available from the FIRE project lead (contact via fire@usz.ch) for researchers who meet the criteria for access to confidential data.

**Funding:** This study was supported financially by AstraZeneca Switzerland. The funders had no role in study design, data collection and analysis, decision to publish, or preparation of the manuscript.

**Competing interests:** LJ reports a speaker honorarium by AstraZeneca. The authors declare that no further conflicts of interest exist.

## Introduction

Chronic kidney disease (CKD) has an estimated global prevalence of approximately 10% and is associated with increased cardiovascular risk and all-cause mortality [1,2]. The role of general practitioners (GPs) in CKD care is important for all disease stages, especially early ones when preventive measures can still avert or delay irreversible end-stage renal disease [3,4]. In Switzerland, data suggest that CKD might affect up to 18% of the patient population in general practice, thus placing a considerable burden on the healthcare system [5].

The *Kidney Disease*: *Improving Global Outcomes* (KDIGO) group published guidelines for evaluation and management of CKD in 2012 [6], which have subsequently been adapted into national recommendations tailored to the general practice setting in several countries including Switzerland [7]. In particular, these guidelines address early detection and staging of CKD based on renal function (RF) assessment using levels of estimated glomerular filtration rate (eGFR) and albuminuria expressed as albumin-to-creatinine ratio (ACR) or urinary albumin concentration (UAC). Moreover, the KDIGO guidelines offer recommendations for pharmacological treatment, therapeutic targets, as well as stage-dependent comprehensiveness of follow-up visits. They also provide guidance about screening for CKD in high-risk populations, especially patients affected by diabetes, hypertension or established cardiovascular disease (eCVD), all of which are frequently encountered in Swiss general practice [8,9].

Various studies have assessed CKD quality of care in general practice by defining and applying quality indicators (QIs) to measure guideline adherence, thereby uncovering dissatisfactory performance in various areas [10–15]. While currently available literature focuses on determinants of QI achievement rates such as the presence of comorbidities or demographics, it does not quantify the variation among GPs. Exploring the extent of unwarranted variation of care would constitute an important first step towards quality improvement [16]. Our study intends to fill this gap by addressing the following aim: To investigate variation and determinants of QI achievement rates for CKD care in Swiss general practice.

## Materials and methods

### Study design, setting and data source

This study followed a retrospective cohort design and used data from the Family medicine Research using Electronic medical records (FIRE) project, a database of anonymised clinical routine data from general practices located in German-speaking Switzerland and hosted by the Institute of Primary Care of the University of Zurich [17]. The FIRE database collects information about patients' and GPs' demographic characteristics, reasons for encounters encoded according to the second revised version of the International Classification of Primary Care (ICPC-2) system [18,19], medication prescriptions as Anatomical Therapeutic Chemical (ATC) codes [20], vital parameters (systolic and diastolic blood pressure), biometric data (body weight, body height), and laboratory test results.

The local Ethics Committee of the Canton of Zurich waived ethical approval for this study, as it lay outside the scope of the Federal Act on Research involving Human Beings (BASEC-Nr. Req-2017-00797) [21]. We follow the *REporting of studies Conducted using Observational Routinely-collected health Data* (RECORD) guidelines for reporting of results [22].

### Objectives and case definitions

Objective of this study was the definition of QIs for CKD care in Swiss general practice with assessment of between-GP variation and determinants of their achievement rates. We defined 14 QIs selected in accordance with data availability within the FIRE project, validation in

existing literature [6,10–15,23], and CKD management guidelines applicable to the study period [6,24–26]. The QIs were grouped into the four categories *Assessment*, *Monitoring*, *Medication*, and *Treatment target achievement*; definitions are provided further below.

We identified CKD cases in terms of laboratory evidence based on RF values in accordance with the KDIGO guidelines. A patient was labelled as affected by CKD upon presence of two eGFR levels < 60 ml/min/1.73m$^2$, and/or two UAC levels $\geq$ 20 mg/l, and/or two ACR levels $\geq$ 3 mg/mmol at least 3 months apart. Where a serum creatinine level was provided in absence of an eGFR value, we estimated the eGFR using the CKD-EPI equation [27]. Due to lacking information on patient ethnicity and with an estimated prevalence of non-Caucasian ethnicity in Swiss general practice near 5% [28], the corresponding factor was neglected. Starting with the earliest measurement in the study period, we staged RF test results into prognostic eGFR (G1–5) or albuminuria (A1–3) categories according to the KDIGO guidelines.

## Cohort definitions

We extracted information from the FIRE database about every patient with at least one consultation recorded during a study period ranging from 1 January 2013 to 31 December 2019 and aged at least 18 years at time of first observation. Patients affected by the predisposing conditions diabetes, hypertension, and eCVD were labelled using criteria outlined in S1 Table. Within this baseline population, we defined two (partially overlapping) cohorts for evaluation of the different QIs:

- **RF assessment cohort** (QIs in the category *Assessment*): All patients observed for a follow-up period of at least 18 months after first evidence of any of the predisposing conditions during the study period.

- **CKD care cohort** (all other QI categories): All patients with laboratory evidence of CKD (as defined in in the previous section) during the study period, excluding patients ever staged G5 (due to likely referral to nephrologists leading to missing follow-up information).

For each QI, we defined a denominator population consisting of patients satisfying specific clinical criteria (e.g., patients with CKD in stages G2–4 or patients with diabetes) within the respective cohort. To be included into the denominator populations for the categories *Assessment* and *Monitoring*, patients required available follow-up periods of at least 18 months. Similarly, available follow-up periods of at least 12 months were required for the category *Medication*. We chose these follow-up lengths to allow for comparison with similar QIs in the literature [10–12]. In the categories *Assessment*, *Monitoring*, and *Medication*, QI achievement was defined as the occurrence of specific diagnostic or therapeutic events during the respective follow-up period. For the category *Treatment target achievement*, QIs were considered achieved if the last respective measurement during the study period lay within a predefined range. Within each denominator population, a numerator population comprised the subset of patients for whom the respective QI was achieved. Table 1 summarizes all QI definitions with denominator and numerator population specifications.

## Statistical analysis

We tabulated variable summaries as counts and percentages or medians with interquartile ranges (IQR) as appropriate. For each QI, we computed an achievement rate as the ratio of numerator to denominator patients.

We explored determinants and GP-level variation of QI achievement rates by means of mixed-effect logistic regression models for achieving the QI in the respective denominator populations. Fixed effects included, wherever applicable, presence of predisposing conditions,

**Table 1. Definition of quality indicators.**

| Category | QI | Numerator | Denominator |
|---|---|---|---|
| Assessment | 1 | Patients receiving RF assessment within a period of 18 months after identification of **diabetes**. | Patients from the RF assessment cohort with a follow-up of at least 18 months following first identification of **diabetes**. |
| | 2 | Patients receiving RF assessment within a period of 18 months after identification of **hypertension**. | Patients from the RF assessment cohort with a follow-up of at least 18 months following first identification of **hypertension**. |
| | 3 | Patients receiving RF assessment within a period of 18 months after identification of **eCVD**. | Patients from the RF assessment cohort with a follow-up of at least 18 months following first identification of **eCVD**. |
| Monitoring | 4 | Patients with at least one assessment of **serum creatinine and/or eGFR** within 18 months from first laboratory evidence of CKD stage **G1–4**. | Patients from the CKD care cohort with a follow-up of at least 18 months following first laboratory evidence of CKD stage **G1–4**. |
| | 5 | Patients with at least one assessment of **albuminuria** within 18 months from first laboratory evidence of CKD stage **G1–4**. | Patients from the CKD care cohort with a follow-up of at least 18 months following first laboratory evidence of CKD stage **G1–4**. |
| | 6 | Patients with at least one assessment of each RF and blood pressure within 18 months from first laboratory evidence of CKD stage **G2–4**. | Patients from the CKD care cohort with a follow-up of at least 18 months following first laboratory evidence of CKD stage **G2–4**. |
| | 7 | Patients with at least one assessment of each RF, blood pressure, CBC and at least one of blood urea and/or electrolytes (at least one of sodium, potassium) within 18 months from first laboratory evidence of CKD stage **G3a–4**. | Patients from the CKD care cohort with a follow-up of at least 18 months following first laboratory evidence of CKD stage **G3a–4**. |
| | 8 | Patients with at least one assessment of each RF, blood pressure, CBC, blood urea and/or electrolytes (at least one of sodium, potassium) and mineral and bone disorder (at least one of blood alkaline phosphatase, calcium, phosphate, parathyroid hormone) within 18 months from first laboratory evidence of CKD stage **G3b–4**. | Patients from the CKD care cohort with a follow-up of at least 18 months following first laboratory evidence of CKD stage **G3b–4**. |
| Medication | 9 | Patients with at least one active **renin-angiotensin-aldosterone system inhibitor** prescription within 12 months following first laboratory evidence of CKD stage **G1–4**. | Patients from the CKD care cohort with a follow-up of at least 12 months following first laboratory evidence of CKD stage **G1–4**. |
| | 10 | Patients aged 50–80 years with at least one active **statin** prescription within 12 months following first laboratory evidence of CKD stage **G1–4**. | Patients from the CKD care cohort aged 50–80 years and with a follow-up of at least 12 months following first laboratory evidence of CKD stage **G1–4**. |
| | 11 | Patients **withheld non-steroidal anti-inflammatory drug** prescription (except acetylsalicylic acid) within 12 months following first laboratory evidence of CKD stage **G2–3b**. | Patients from the CKD care cohort with a follow-up of at least 12 months following first laboratory evidence of CKD stage **G2–3b**. |
| Treatment target achievement | 12 | Patients with laboratory evidence of CKD stage **G1–4** and latest **blood pressure** measurement below 140/90 mmHg. | Patients from the CKD care cohort with at least one blood pressure measurement after first laboratory evidence of CKD stage **G1–4**. |
| | 13 | Patients with laboratory evidence of CKD stage **G1–4** and **diabetes** and latest **blood pressure** measurement below 130/80 mmHg. | Patients from the CKD care cohort with at least one blood pressure measurement after first laboratory evidence of CKD stage **G1–4** and **diabetes**. |
| | 14 | Patients with laboratory evidence of CKD stage **G1–4** and latest **body mass index** in the range 20–25. | Patients from the CKD care cohort with at least one body mass index measurement after first laboratory evidence of CKD stage **G1–4**. |

See main text for definition of laboratory evidence of chronic kidney disease (CKD). Renal function (RF) assessments are defined as measurement of serum creatinine and/or albuminuria. Abbreviations: eCVD, established cardiovascular disease; eGFR, estimated glomerular filtration rate; QI, quality indicator.

patient and GP demographics, and urbanity of practice location (according to the Swiss Federal Statistical Office [29]). GP-level random intercepts were used to account for clustering of patients within their respective GPs. We report fixed-effect summaries of regression models in terms of odds ratios (OR) with corresponding 95% confidence intervals (CI) and display them graphically as forest plots.

GP-level variation of QI achievement rates is summarised in terms of intraclass correlation coefficients (ICCs, [30]) and of range odds ratios (rOR) across the central 90% of fitted random intercepts. Both were determined in analogous mixed-effect models under omission of GP-level fixed effects, where the influence of individual GPs was entirely expressed by the

corresponding random intercept. We computed the rOR as $e^{q95 - q05}$, where q95 and q05 denote the empirical 95th and 5th percentile of predicted random intercepts, respectively, and e is Euler's number. While the ICC can be regarded as a proportion of variation attributable to the GP, the rOR provides a measure of differences across GPs on the same scale and with a similar interpretation as fixed-effect ORs. Use of the central 90% instead of the full range of fitted random intercepts reduces emphasis on outliers.

All statistical analyses were conducted with the statistical software R version 4.1.0 (R Foundation for Statistical Computing, Vienna, Austria) [31]; the library *lme4* was used for mixed-effect model fitting [32]. We handled missing demographic data by pairwise deletion for descriptive and by listwise deletion for regression analyses.

## Results

### Cohort characteristics

In total, 578,802 patients (age median 46 years, IQR 31–61 years, 52.7% female) from 483 GPs (age median 49 years, IQR 41–57 years, 43.9% female) in 182 practices were included into the baseline population and approached for eligibility. Out of these, 47,201 patients met the criteria for the RF assessment cohort (age median 68 years, IQR 57–77 years, 48.7% female) and 14,627 patients for the CKD care cohort (age median 80 years, IQR 73–86 years, 57.5% female). Within the latter, the most common eGFR stage at cohort inclusion was G3a (53.8%), followed by G3b (28.3%). Cohort characteristics are summarised in Table 2; Fig 1 displays the cohort selection process.

**Table 2. Study participant characteristics.**

| General practitioners (*n* = 483) | Value | Missing, *n* (%) |
|---|---|---|
| Gender, *n* (%) | | 6 (1.2) |
| Female | 212 (43.9) | |
| Male | 265 (54.9) | |
| Age at first observation in years, *n* (%) | | 23 (4.8) |
| < 45 | 182 (37.7) | |
| 45–60 | 216 (44.7) | |
| ≥ 60 | 62 (12.8) | |
| Urban practice location, *n* (%) | 361 (74.7) | 0 (0.0) |
| **Patients: renal function assessment cohort (*n* = 47,201)** | | **Missing, *n* (%)** |
| Gender, *n* (%) | | 0 (0.0) |
| Female | 22,988 (48.7) | |
| Male | 24,210 (51.3) | |
| Other* | 3 (0.0) | |
| Age at inclusion in years, *n* (%) | | 0 (0.0) |
| < 40 | 1,922 (4.1) | |
| 40–59 | 12,454 (26.4) | |
| 60–79 | 23,929 (50.7) | |
| ≥ 80 | 8,896 (18.8) | |
| Predisposing condition, *n* (%) | | |
| Diabetes | 13,625 (28.9) | |
| Hypertension | 40,304 (85.4) | |
| eCVD | 16,697 (35.4) | |

*(Continued)*

**Table 2.** (Continued)

| General practitioners (*n* = 483) | Value | Missing, *n* (%) |
|---|---|---|
| Patients: chronic kidney disease care cohort (*n* = 14,627) | | Missing, *n* (%) |
| Gender, *n* (%) | | 0 (0.0) |
| Female | 8,420 (57.6) | |
| Male | 6,204 (42.4) | |
| Other* | 3 (0.0) | |
| Age at inclusion in years, *n* (%) | 80 (73–86) | 0 (0.0) |
| < 65 | 1251 (8.6) | |
| 65–79 | 5514 (37.7) | |
| ≥ 80 | 7862 (53.7) | |
| Predisposing condition, *n* (%) | | |
| Diabetes | 3,642 (24.9) | |
| Hypertension | 8,491 (57.9) | |
| eCVD | 3,999 (27.3) | |
| None of the above | 4,368 (29.8) | |
| G stage at cohort inclusion (eGFR range in ml/min/1.73m$^2$), *n* (%) | | |
| G1 (≥ 90) | 217 (1.4) | |
| G2 (60–89) | 627 (4.2) | |
| G3a (45–59) | 8,105 (53.8) | |
| G3b (30–44) | 4,265 (28.3) | |
| G4 (15–29) | 1,251 (8.3) | |
| A stage at cohort inclusion (ACR range in mg/mmol), *n* (%) | | |
| A1 (< 3) | 266 (1.8) | |
| A2 (3–30) | 715 (4.8) | |
| A3 (> 30) | 81 (0.5) | |
| Laboratory values (latest), median (IQR); *n* (%) | | |
| Serum creatinine, μmol/l | 107 (89–129); 14,530 (99.3) | |
| eGFR, ml/min/1.73 m$^2$ | 48 (38–57); 14,530 (99.3) | |
| ACR, mg/mmol | 4.5 (1.5–14.0); 2,384 (16.3) | |
| Urinary albumin concentration, mg/l | 32.8 (10.4–100.0); 2,294 (15.7) | |
| Hemoglobin, g/l | 130 (119–141); 10,609 (72.4) | |
| Sodium, mmol/l | 141 (139–143); 4,955 (33.8) | |
| Potassium, mmol/l | 4.3 (4.0–4.6); 6,691 (45.7) | |
| Urea, mmol/l | 9.9 (7.8–13.2); 825 (5.6) | |
| Alkaline phosphatase, U/l | 75 (60–97); 1,785 (12.2) | |
| Calcium, mmol/l | 2.36 (2.28–2.46); 826 (5.6) | |
| Phosphate, mmol/l | 1.12 (0.99–1.24); 158 (1.1) | |
| Parathyroid hormone, ng/l | 63.9 (38.1–97.1); 77 (0.5) | |
| Glycated hemoglobin, % | 6.1 (5.6–7.0); 5,723 (39.1) | |
| Low-density lipoprotein, mmol/l | 2.58 (1.90–3.36); 2,025 (13.8) | |
| Total cholesterol, mmol/l | 4.79 (4.00–5.73); 3,107 (21.2) | |
| Physical examination (latest), median (IQR); *n* (%) | | |
| Systolic blood pressure, mmHg | 135 (122–150); 13,572 (92.8) | |
| Diastolic blood pressure, mmHg | 77 (70–84); 13,559 (92.7) | |
| Body mass index, kg/m$^2$ | 27 (24–31); 8,770 (60.0) | |

Abbreviations: ACR, albumin-to-creatinine ratio; eCVD, established cardiovascular disease; eGFR, estimated glomerular filtration rate; IQR, interquartile range.

*Due to their rare occurrence, we omitted patients in this gender category from regression analyses.

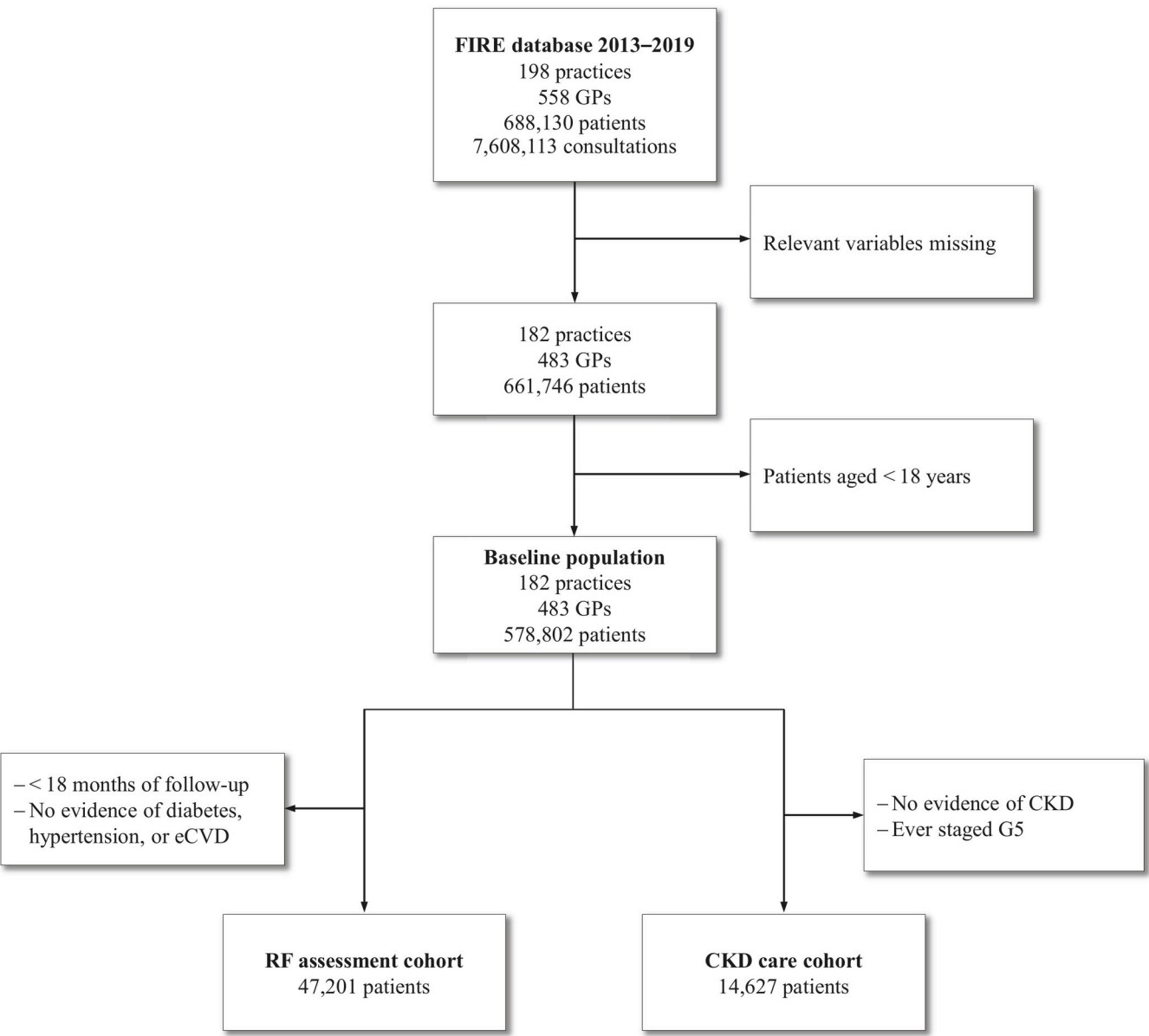

**Fig 1. Study flowchart.** Selection of the renal function (RF) assessment cohort and the chronic kidney disease (CKD) care cohort. Abbreviations: eCVD, established cardiovascular disease; GP, general practitioner.

### QI achievement and variation

Table 3 summarises QI achievement rates as well as corresponding ICCs and rORs determined from regression models, while the determinants of achievement rates are displayed in the forest plots of Fig 2. Detailed numerical results are provided in S2–S5 Tables.

We found the highest achievement rate for withholding non-steroidal anti-inflammatory drug (NSAID) prescriptions in patients in stages G2–G3b (82.6%). The lowest was observed for albuminuria monitoring at 18.1%, which was paired with comparatively high GP-level variation (rOR 16.5, ICC 0.28). The highest levels of between-GP variation were found for RF assessment in patients with diabetes (rOR 26.5, ICC 0.31), eCVD (rOR 17.4, ICC 0.28), and hypertension (rOR 17.2, ICC 0.24). Male patients were more likely to meet QIs concerning RF

**Table 3. Rates and variation of quality indicator achievement.**

| Category | QI | Description | Achievement rate, % | rOR | ICC |
|---|---|---|---|---|---|
| Assessment | 1 | RF; diabetes (18) | 76.2 | 26.5 | 0.31 |
| | 2 | RF; hypertension (18) | 66.7 | 17.2 | 0.24 |
| | 3 | RF; eCVD (18) | 69.5 | 17.4 | 0.28 |
| Monitoring | 4 | eGFR/SCr; G1–4 (18) | 70.0 | 4.1 | 0.09 |
| | 5 | ACR/UAC; G1–4 (18) | 18.1 | 16.5 | 0.28 |
| | 6 | RF+BP; G2–4 (18) | 56.7 | 11.2 | 0.19 |
| | 7 | RF+BP+CBC+chem; G3a–4 (18) | 56.2 | 8.4 | 0.17 |
| | 8 | RF+BP+CBC+chem+MBD; G3b–4 (18) | 45.3 | 3.6 | 0.12 |
| Medication | 9 | RAAS inhibitor; G1–4 (12) | 70.7 | 3.2 | 0.07 |
| | 10 | Statin; G1–4, 50–80 years (12) | 49.8 | 2.7 | 0.06 |
| | 11 | Withheld NSAID; G2–3b (12) | 82.6 | 3.2 | 0.07 |
| Treatment target achievement | 12 | Latest BP < 140/90 mmHg; G1–4 | 54.9 | 2.6 | 0.06 |
| | 13 | Latest BP < 130/80 mmHg; G1–4, diabetes | 54.0 | 3.2 | 0.09 |
| | 14 | Latest BMI 20–25 kg/m$^2$; G1–4 | 33.3 | 1.5 | 0.02 |

Range odds ratios (rOR) and intraclass correlation coefficients (ICC) are determined from the regression models. The column "Description" gives a shorthand outline of the respective QI in the format "outcome; denominator population (follow-up period in months)" (see Table 1 for detailed descriptions). Note that the denominator populations for the category *Assessment* encompass all patients with the respective predisposing condition, not only those affected by chronic kidney disease. Abbreviations: ACR, albumin-to-creatinine ratio; BMI, body mass index; BP, blood pressure; CBC, complete blood count; chem, blood chemistry (at least one of urea, sodium, potassium); eCVD, established cardiovascular disease; MBD, mineral and bone disorder screening; NSAID, non-steroidal anti-inflammatory drug; RAAS, renin-angiotensin-aldosterone system; renal function; SCr, serum creatinine; UAC urinary albumin concentration.

assessment in presence of diabetes or hypertension, albuminuria monitoring, RAAS inhibitor prescription, statin prescription, withholding NSAID prescription, and blood pressure targets. Patient age showed positive association with achievement of RF assessment in presence of predisposing conditions and negative association with achievement of albuminuria monitoring and blood pressure targets.

## Discussion

Evidence on GP-level variation of quality of CKD care is scarce. In this study, we defined and evaluated achievement rates of 14 QIs for care of over 47,000 patients at risk for CKD and of over 14,000 patients with CKD in Swiss general practice, quantified their GP-level variation, and investigated their determinants. We revealed relevant gaps in terms of low achievement rates and high GP-level variation, most notably with respect to albuminuria monitoring and RF assessment in patients with predisposing conditions. In addition, we exposed associations of QI achievement with patient characteristics that may point at disparities in CKD care.

We identified CKD patients by means of the KDIGO laboratory definition and obtained a cohort aged 80 years in median with 58% women, a demographic close to that of similar studies [11,12]. The advanced age of CKD patients in general practice contrasts with the study populations of most clinical trials addressing advanced CKD [33], which poses an important challenge to guideline applicability. Prevalence of multimorbidity is high among elderly patients [34], and strict adherence to guidelines may not always be appropriate for them. In addition, GPs may consider eGFR levels below 60 ml/min/1.73 m$^2$ in aged patients to reflect physiological decline of RF rather than to constitute a risk factor for end-stage renal disease. Indeed, the concern of overdiagnosis of CKD in the elderly has been raised [35], and definitions using age-adjusted thresholds of eGFR have been proposed [36].

Our analysis expressed GP-level variation in terms of rORs and ICCs from regression models. To the best of our knowledge, this is the first study to provide a comprehensive overview of these quantities for QIs in the context of CKD care. It thus enables direct comparisons of practice variation across its different domains and sets starting points for planning and evaluation of future interventions. Overall, we found marked GP-level variation in achievement of most of the examined QIs, with rORs often surpassing fixed-effect ORs of the respective models by several units. In addition, almost all observed ICCs were higher than 0.05, which is rather unusual for processes in general practice and may express unwarranted variation [37].

The most striking finding, characterised by high variation combined with the lowest achievement rate, concerned albuminuria monitoring in patients with established CKD. Our results revealed even lower albuminuria testing rates than studies from Australia [11], the Netherlands [38] and Canada [10,14]. The difference may in part be explained by inclusion of patients in all CKD stages into our cohort, as opposed to restriction to stages G3 or higher in most of these studies. Nevertheless, we reproduced a known association of albuminuria assessment with male gender and diabetes, as well as lower testing rates with increasing age [10,38,39] and a contrast to high achievement rates regarding serum creatinine testing [12]. The relatively high proportion of patients with RAAS inhibitor prescriptions (over 70% during 12 months follow-up) might play an important role, as GPs may deem albuminuria measurement not indicated in this population. While evidence for a clear benefit of albuminuria monitoring in these patients is scarce, to the point that the American College of Physicians has even recommended against it [40], this view has been subject to controversy [41].

While achievement rates of QIs addressing RF assessment in patients at risk for CKD were generally high, the high rOR and ICC values could indicate unwarranted practice variation. In addition, the marked association of RF testing with increasing age is worrisome, as it may reveal lost opportunities to prevent CKD progression in early stages by underuse of screening. Stage-appropriate completeness of monitoring within 18 months follow-up showed moderate achievement rates around 56% when considering blood pressure, anaemia screening and blood chemistry, but dropped to 45% when involving screening for mineral and bone disorder. However, GPs may refer a certain proportion of patients in the rather advanced stages G3b–4 considered for this QI is to nephrologists who take charge of these analyses.

The best-met of all QIs involved withholding NSAIDs to CKD patients and showed an achievement rate over 80%. This proportion aligns with the recently determined yearly prescription rates of NSAIDs in the Swiss general practice population of the same age range [42]. Whether GPs take presence of CKD into account when prescribing NSAIDs therefore remains undetermined. Nonetheless, GP-level variation was among the lowest with an ICC of 0.07, which lies below a value of 0.14 found for general high-risk NSAID initiation in general practice [43].

Management of blood pressure resulted in a moderate target achievement of 55%. The relatively low ICC of 0.06, only slightly higher than in a British study [15], suggests a higher dependence of blood pressure adjustment on patient- rather than GP-level factors. In this context, it is important to take the advanced age of the CKD care cohort into account, as concerns of frailty and polypharmacy may lead to GPs opting for less intensive blood pressure adjustment. While new evidence supporting intensive blood pressure control also in the elderly has been integrated into the 2021 update of the KDIGO guidelines [44,45], the observed behaviour was in accordance with recommendations pertinent to the study period [46]. On the other hand, the comparatively low prescription rates of cholesterol-lowering drugs in CKD patients aged 50–80 years were more concerning, and may reflect a generally prevalent undertreatment with statins found in Swiss general practice [47].

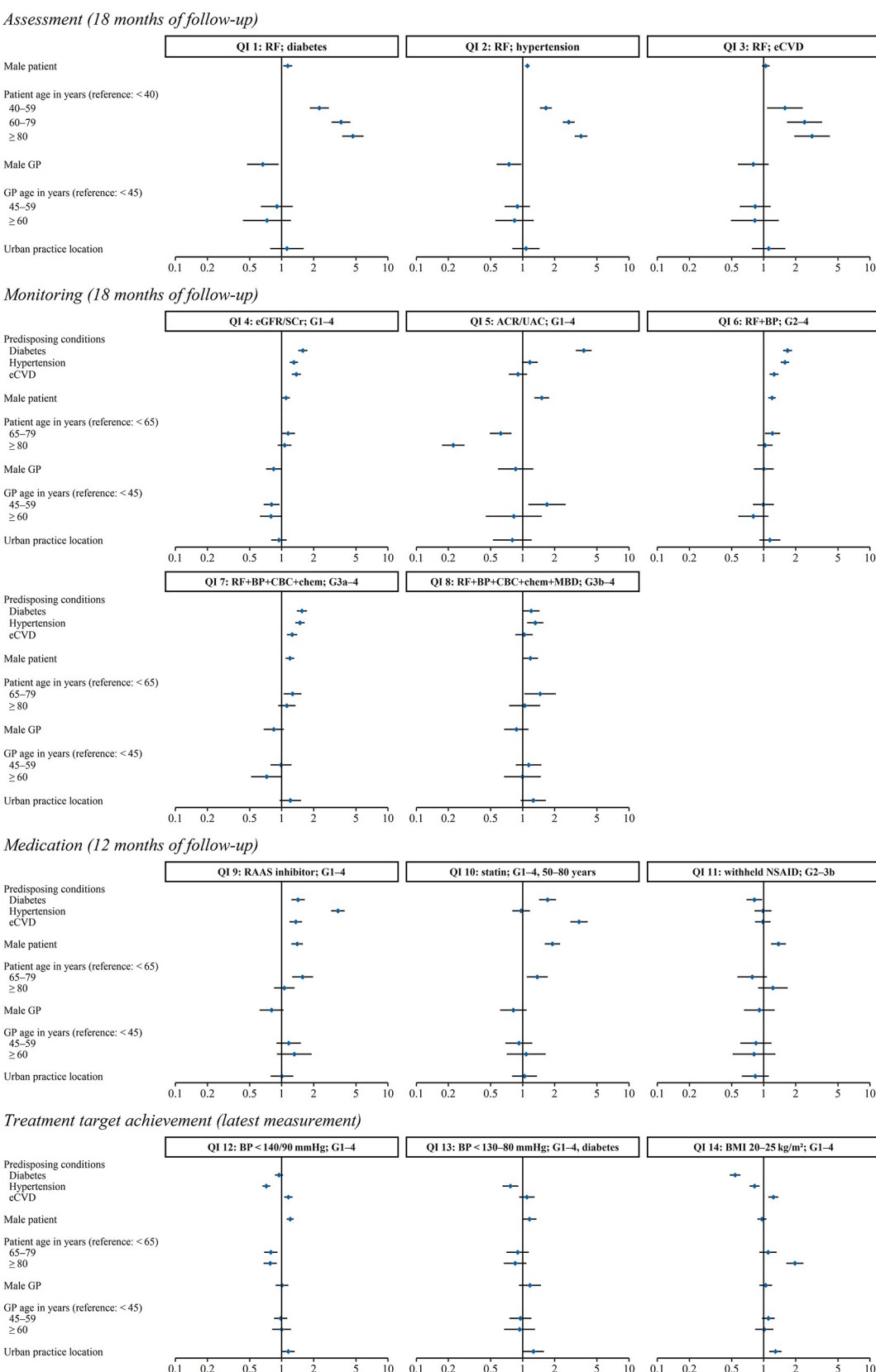

*Assessment (18 months of follow-up)*

*Monitoring (18 months of follow-up)*

*Medication (12 months of follow-up)*

*Treatment target achievement (latest measurement)*

**Fig 2. Determinants of quality indicator achievement.** These forest plots display effect sizes expressed as odds ratios (OR) with 95% confidence intervals (CI), grouped by quality indicator (QI) category. Next to each QI number is a shorthand outline in the format "outcome; denominator population" (see Table 1 for detailed descriptions). Note that the denominator populations for the category *Assessment* encompass all patients with the respective predisposing condition, not only those affected by chronic kidney disease. Abbreviations: ACR, albumin-to-creatinine ratio; BMI, body mass index, BP, blood pressure; CBC, complete blood count; chem, blood chemistry (at least one of urea, sodium, potassium); eCVD, established cardiovascular disease; MBD, mineral and bone disorder screening; NSAID, non-steroidal anti-inflammatory drug; RAAS, renin-angiotensin-aldosterone system; RF, renal function; SCr, serum creatinine; UAC, urinary albumin concentration.

Common to QIs involving management of cardiovascular risk factors in CKD was association of achievement with male gender. This finding aligns with a growing body of evidence pointing at a gender gap with respect to management of modifiable cardiovascular risk factors in general practice [48,49]. Apart from a Canadian study [10], we are not aware of literature that exposed gender differences in the context of CKD care.

Interestingly, the presence of predisposing conditions, especially diabetes, was often associated with higher QI achievement. This finding aligns with a corresponding association described in the context of cardiovascular risk management [50] and may mirror a possible perception of CKD care being only relevant in the presence of comorbidities.

The observed associations of GP-level characteristics and practice location with QI achievement were generally non-significant. Still, the results suggest that rather weak associations may be compatible with the observed data, and larger samples of GPs may be needed to achieve statistical significance. A similar situation holds for practice location: While analyses conducted in Canada [10,51], the Netherlands [38], and Australia [11,52] have revealed various urban-rural disparities, we were generally not able to observe relevant patterns. Notable exceptions were higher QI achievement rates of blood pressure and body mass index targets in urban locations. The former have not been observed previously [53], but the latter may be explained by known prevalence patterns of overweight and obesity in Switzerland [54].

## Strengths and limitations

A major strength of our study is the focus on routine clinical data, which reflects processes of care in real-life general practice. In particular, it provides direct insight into the alleged basis of decision making in form of results from vital parameter and laboratory test measurements. Access to these parameters constitutes a major asset in comparison with other sources such as healthcare administrative data.

An important limitation was imposed by the nature of data captured by the FIRE database, since information on referrals to nephrologists, dialysis, transplantation, and mortality would have substantially contributed to a more exhaustive assessment. Moreover, we were only able to identify CKD by means of its laboratory definition, as we had no access to diagnostic documentation in form of free-text entries or of diagnostic codes. Consequently, we may have missed several patients in our analyses, especially in early stages of disease. However, this proportion may be negligible, since studies have revealed that GPs document CKD in patient records at relatively low frequency [11,14].

A substantial proportion of Swiss GPs did not make use of electronic medical records during the study period [55], and voluntary participation in the FIRE project may have led to selection bias in favour of higher quality of care. These aspects may have impaired representativeness of the Swiss GP population.

Lastly, data quality issues can lead to failed capture of important information from electronic records, resulting in underestimation of QI achievement rates. Nonetheless, we aimed to minimize this source of bias by restricting on practices exporting data at sufficiently high quality of the variables relevant to assessment of each QI individually.

## Conclusion

In summary, these findings suggest potentially unwarranted variation within specific areas of CKD care in Swiss general practice. Particular challenges involve monitoring of albuminuria, screening for CKD in presence of predisposing conditions, care of CKD patients not affected by comorbidities, and care of female patients with CKD. Our study highlights the need for educational efforts aimed at reducing the observed variation. Corresponding initiatives would need to take age and morbidity of the general practice CKD population into account to ensure an appropriate implementation of clinical practice guidelines. Routine clinical databases may play an important role to evaluate the effects of such interventions.

## Supporting information

**S1 Table. Operationalized criteria for identification of conditions and events.** Criteria were defined using International Classification of Primary Care (ICPC-2) codes, Anatomical Therapeutic Chemical (ATC) codes, vital parameters, and laboratory values.
(PDF)

**S2 Table. Determinants of quality indicator achievement in the category** *Assessment.* "Full model" denotes the regression model including all predictors, while "Null model" denotes the model where demographic characteristics of general practitioners (GPs) were omitted. Abbreviations: CI, confidence interval; ICC, intraclass correlation coefficient; OR, odds ratio; QI, quality indicator.
(PDF)

**S3 Table. Determinants of quality indicator achievement in the category** *Monitoring.* "Full model" denotes the regression model including all predictors, while "Null model" denotes the model where demographic characteristics of general practitioners (GPs) were omitted. Abbreviations: CI, confidence interval; eCVD, established cardiovascular disease; ICC, intraclass correlation coefficient; OR, odds ratio; QI, quality indicator.
(PDF)

**S4 Table. Determinants of quality indicator achievement in the category** *Medication.* "Full model" denotes the regression model including all predictors, while "Null model" denotes the model where demographic characteristics of general practitioners (GPs) were omitted. Abbreviations: CI, confidence interval; eCVD, established cardiovascular disease; ICC, intraclass correlation coefficient; OR, odds ratio; QI, quality indicator.
(PDF)

**S5 Table. Determinants of quality indicator achievement in the category** *Treatment target achievement.* "Full model" denotes the regression model including all predictors, while "Null model" denotes the model where demographic characteristics of general practitioners (GPs) were omitted. Abbreviations: CI, confidence interval; eCVD, established cardiovascular disease; ICC, intraclass correlation coefficient; OR, odds ratio; QI, quality indicator.
(PDF)

**S1 Dataset. Data of the renal function assessment cohort.**
(CSV)

**S2 Dataset. Data of the chronic kidney disease care cohort.**
(CSV)

## Acknowledgments

The authors thank Fabio Valeri for management of the FIRE database and the FIRE study group of general practitioners for contribution of data to this study.

## Author Contributions

**Conceptualization:** Levy Jäger, Stefan Markun.

**Data curation:** Levy Jäger.

**Formal analysis:** Levy Jäger.

**Funding acquisition:** Thomas Rosemann.

**Investigation:** Levy Jäger.

**Methodology:** Levy Jäger, Stefan Markun.

**Project administration:** Thomas Rosemann, Jakob Martin Burgstaller.

**Resources:** Thomas Rosemann, Jakob Martin Burgstaller, Oliver Senn.

**Software:** Levy Jäger.

**Supervision:** Oliver Senn, Stefan Markun.

**Validation:** Levy Jäger.

**Visualization:** Levy Jäger.

**Writing – original draft:** Levy Jäger.

**Writing – review & editing:** Levy Jäger, Thomas Rosemann, Jakob Martin Burgstaller, Oliver Senn, Stefan Markun.

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
