## [Decision Letter · Decision Letter 0]

9 May 2022

PONE-D-22-07326Quality and variation of care for chronic kidney disease in Swiss general practice: A retrospective database studyPLOS ONE

Dear Dr. Jäger,

Thank you for submitting your manuscript to PLOS ONE. After careful consideration, we feel that it has merit but does not fully meet PLOS ONE’s publication criteria as it currently stands. Therefore, we invite you to submit a revised version of the manuscript that addresses the points raised during the review process. We appreciate the interesting and well conducted study. However, there are some minor points raised by the reviewer #2. Please carefully respond to the reviewer comments and suggestions.

We look forward to receiving your revised manuscript.

Kind regards,

Vipa Thanachartwet, M.D.

Academic Editor

PLOS ONE

Journal Requirements:

Reviewers' comments:

Reviewer's Responses to Questions

**Comments to the Author**

1. Is the manuscript technically sound, and do the data support the conclusions?

Reviewer #1: Yes

Reviewer #2: Yes

2. Has the statistical analysis been performed appropriately and rigorously? 

Reviewer #1: Yes

Reviewer #2: Yes

3. Have the authors made all data underlying the findings in their manuscript fully available?

Reviewer #1: Yes

Reviewer #2: Yes

4. Is the manuscript presented in an intelligible fashion and written in standard English?

Reviewer #1: Yes

Reviewer #2: Yes

5. Review Comments to the Author

Reviewer #1: It is a very interesting study that has allowed to explore the implementation of suggested quality indicators in different clinical practice guidelines by primary care doctors. Despite reporting in this study, acceptable rates for the evaluation of renal function (around 70%) in patients with predisposing conditions, the ideal would be to evaluate 100% of this population. As they raise in the discussion, it is exploration is greater in older patients, that is, we do not study patients who can really favor an early diagnosis, the population with ages between 40 and 59 years. On the other hand, the determination of albuminuria is scarce in the oldest population.

The findings of this study are similar to our medical practice in Colombia where we find that 30 to 35% of the population with arterial hypertension and diabetes mellitus is not determined by renal function.

Reviewer #2: Thanks for asking me to review this well conducted retrospective study of large cohort of patients with renal failure/CKD based on laboratory parameters seen in General Practice. Two main strengths of the study are the size of the cohort and length of study period, although retrospective.

I guess this study highlighted the discordance between international and national guidelines and the real world general practice in screening, identification, assessment and management of CKD patients.

My question to the authors is:

Have you identified any GP based factors for this variation of care? in relation to age, gender, urban vs rural.

Are there any focussed educational programs for GPs to access in terms of CKD management?

What are your recommendations to address this gap of care and variation in practice?

Can you predict reasons for why NSAID discontinuation in CKD has the highest priority for GPs and are there any lessons to learn from that to apply for other situations like statin use etc..?

6. PLOS authors have the option to publish the peer review history of their article (what does this mean?). If published, this will include your full peer review and any attached files.

Reviewer #1: No

Reviewer #2: **Yes: **Sree Krishna Venuthurupalli

---

## [Author Response · Author response to Decision Letter 0]

13 Jul 2022

Response to Reviewer 1

Reviewer’s comments: It is a very interesting study that has allowed to explore the implementation of suggested quality indicators in different clinical practice guidelines by primary care doctors. Despite reporting in this study, acceptable rates for the evaluation of renal function (around 70%) in patients with predisposing conditions, the ideal would be to evaluate 100% of this population. As they raise in the discussion, it is exploration is greater in older patients, that is, we do not study patients who can really favor an early diagnosis, the population with ages between 40 and 59 years. On the other hand, the determination of albuminuria is scarce in the oldest population.

The findings of this study are similar to our medical practice in Colombia where we find that 30 to 35% of the population with arterial hypertension and diabetes mellitus is not determined by renal function.

Authors’ reply: We thank the reviewer for appreciating our work and for the encouraging comments. We agree that while achievement rates of QIs addressing renal function assessment in patients with predisposing conditions were comparatively high, they are still not ideal. We hope that results of our study may be useful for informing initiatives aimed at reducing the observed variation in renal function assessment and at increasing screening rates among patients at high risk for chronic kidney disease.

Response to Reviewer 2

Reviewer’s general remarks: Thanks for asking me to review this well conducted retrospective study of large cohort of patients with renal failure/CKD based on laboratory parameters seen in General Practice. Two main strengths of the study are the size of the cohort and length of study period, although retrospective.

I guess this study highlighted the discordance between international and national guidelines and the real world general practice in screening, identification, assessment and management of CKD patients.

My question to the authors is:

Authors’ general reply: We thank the reviewer for appreciating our study and the constructive remarks. We would like to address the questions point by point. Page and line numbers refer to the revised version of the manuscript.

Reviewer’s question 1: Have you identified any GP based factors for this variation of care? in relation to age, gender, urban vs rural.

Authors’ reply 1: In our study, we failed to identify any relevant and significant associations between GP-level characteristics and QI achievement. Urbanity of practice location played a role only for QIs involving treatment target achievement. We have elaborated these points in the corresponding paragraph of the discussion as follows:

Pages 14¬¬–15, lines 281–290: The observed associations of GP-level characteristics and practice location with QI achievement were generally non-significant. Still, the results suggest that rather weak associations may be compatible with the observed data, and larger samples of GPs may be needed to achieve statistical significance. A similar situation holds for practice location: While analyses conducted in Canada [1, 2], the Netherlands [3], and Australia [4, 5] have revealed various urban-rural disparities, we were generally not able to observe relevant patterns. Notable exceptions were higher QI achievement rates of blood pressure and body mass index targets in urban locations. The former have not been observed previously [6], but the latter may be explained at least in part by known prevalence patterns of overweight and obesity in Switzerland [7].

Reviewer’s question 2: Are there any focussed educational programs for GPs to access in terms of CKD management?

Authors’ reply 2: We are not aware of any focused or systematic large-scale educational programs for GPs on the topic of CKD care in Switzerland. In our experience, educational efforts are limited to occasional presentations in national congresses and training sessions by nephrology specialists. Our study has stressed the need for more targeted educational interventions, which we have included in the following sentence at the end of the conclusion (see also our reply to question 3):

Page 16, lines 318–321: Our study highlights the need for educational efforts aimed at reducing the observed variation. Corresponding initiatives would need to take age and morbidity of the general practice CKD population into account to ensure an appropriate implementation of clinical practice guidelines.

Reviewer’s question 3: What are your recommendations to address this gap of care and variation in practice?

Authors’ reply 3: As mentioned in our reply to the previous point, we believe that the observed gaps in care may be addressed by targeted educational efforts to raise awareness of the importance of CKD care. In particular, we have provided a description of the CKD population encountered by Swiss GPs, revealing its advanced age and morbidity, which would play an important role for informing corresponding initiatives. We have included this idea to strengthen the conclusions as follows (see also our reply to question 2):

Page 16, lines 318–321: Our study highlights the need for educational efforts aimed at reducing the observed variation. Corresponding initiatives would need to take age and morbidity of the general practice CKD population into account to ensure an appropriate implementation of clinical practice guidelines.

In addition, our study is the first to provide a comprehensive assessment of CKD care in Swiss general practice, and our data source may serve as a starting point to establish a structured data monitoring tool where the implementation of educational interventions may be evaluated. This aspect is now mentioned in the conclusions:

Page 16, lines 321–322: Routine clinical databases may play an important role to evaluate the effects of such interventions.

Reviewer’s question 4: Can you predict reasons for why NSAID discontinuation in CKD has the highest priority for GPs and are there any lessons to learn from that to apply for other situations like statin use etc..?

Authors’ reply 4: We thank the author for raising an important point concerning discontinuation and withholding prescribing of potentially inappropriate medication in general practice. We believe that in Switzerland, a high awareness of NSAID-associated risks may be due to the popularity of lists and recommendations concerning potentially inappropriate prescribing for patients aged over 65 years such as the German PRISCUS list [8] or the American Society of Geriatrics Beers Criteria® [9]. Since this aspect is rather speculative, we have refrained from including a corresponding sentence in the manuscript. However, the point of the high rate of NSAID withholding possibly not rooting in CKD specifically, but rather reflecting generally low prescribing rates in the corresponding age group (Page13, lines 257–259), is raised. 

These considerations do not apply to the use of statins among the younger sub-population of CKD patients aged 50–80 years as investigated in QI 10. We think that the comparatively low achievement rate may be addressed in educational interventions to raise the awareness of cardiovascular risk management in CKD patients (especially in light of the observed gender gap).

References

1. Bello AK, Hemmelgarn B, Lin M, Manns B, Klarenbach S, Thompson S, et al. Impact of remote location on quality care delivery and relationships to adverse health outcomes in patients with diabetes and chronic kidney disease. Nephrol Dial Transplant. 2012;27(10):3849-55. Epub 2012/07/05. doi: 10.1093/ndt/gfs267. PubMed PMID: 22759385.

2. Bello AK, Ronksley PE, Tangri N, Kurzawa J, Osman MA, Singer A, et al. Quality of Chronic Kidney Disease Management in Canadian Primary Care. JAMA Netw Open. 2019;2(9):e1910704. Epub 2019/09/05. doi: 10.1001/jamanetworkopen.2019.10704. PubMed PMID: 31483474; PubMed Central PMCID: PMC6727682.

3. Van Gelder VA, Scherpbier-De Haan ND, De Grauw WJ, Vervoort GM, Van Weel C, Biermans MC, et al. Quality of chronic kidney disease management in primary care: a retrospective study. Scand J Prim Health Care. 2016;34(1):73-80. Epub 2016/02/09. doi: 10.3109/02813432.2015.1132885. PubMed PMID: 26853071; PubMed Central PMCID: PMC4911031.

4. Khanam MA, Kitsos A, Stankovich J, Kinsman L, Wimmer B, Castelino R, et al. Chronic kidney disease monitoring in Australian general practice. Aust J Gen Pract. 2019;48(3):132-7. Epub 2019/07/01. doi: 10.31128/ajgp-07-18-4630. PubMed PMID: 31256479.

5. Bezabhe WM, Kitsos A, Saunder T, Peterson GM, Bereznicki LR, Wimmer BC, et al. Medication Prescribing Quality in Australian Primary Care Patients with Chronic Kidney Disease. J Clin Med. 2020;9(3). Epub 2020/03/19. doi: 10.3390/jcm9030783. PubMed PMID: 32183127; PubMed Central PMCID: PMC7141290.

6. Walther D, Curjuric I, Dratva J, Schaffner E, Quinto C, Rochat T, et al. High blood pressure: prevalence and adherence to guidelines in a population-based cohort. Swiss Med Wkly. 2016;146:w14323. Epub 20160711. doi: 10.4414/smw.2016.14323. PubMed PMID: 27399797.

7. Swiss Federal Statistical Office. Schweizerische Gesundheitsbefragung 2017: Übergewicht und Adipositas. Neuchâtel, Switzerland2020 [cited 2022 Jun 13]. Available from: https://www.bfs.admin.ch/bfs/de/home/statistiken/gesundheit/erhebungen/sgb.assetdetail.14147705.html.

8. Holt S, Schmiedl S, Thürmann PA. Potentially inappropriate medications in the elderly: the PRISCUS list. Dtsch Arztebl Int. 2010;107(31-32):543.

9. American Geriatrics Society Beers Criteria® Update Expert Panel. American Geriatrics Society 2019 Updated AGS Beers Criteria® for Potentially Inappropriate Medication Use in Older Adults. J Am Geriatr Soc. 2019;67(4):674-94. doi: https://doi.org/10.1111/jgs.15767.

---

## [Decision Letter · Decision Letter 1]

25 Jul 2022

Quality and variation of care for chronic kidney disease in Swiss general practice: A retrospective database study

PONE-D-22-07326R1

Dear Dr. Levy Jäger,

We’re pleased to inform you that your manuscript has been judged scientifically suitable for publication and will be formally accepted for publication once it meets all outstanding technical requirements.

Kind regards,

Vipa Thanachartwet, M.D.

Academic Editor

PLOS ONE

Additional Editor Comments (optional):

All issues were addressed according to the reviewers' comments and suggestions. 

Reviewers' comments:

Reviewer's Responses to Questions

**Comments to the Author**

1. If the authors have adequately addressed your comments raised in a previous round of review and you feel that this manuscript is now acceptable for publication, you may indicate that here to bypass the “Comments to the Author” section, enter your conflict of interest statement in the “Confidential to Editor” section, and submit your "Accept" recommendation.

Reviewer #1: (No Response)

2. Is the manuscript technically sound, and do the data support the conclusions?

Reviewer #1: (No Response)

3. Has the statistical analysis been performed appropriately and rigorously? 

Reviewer #1: (No Response)

4. Have the authors made all data underlying the findings in their manuscript fully available?

Reviewer #1: (No Response)

5. Is the manuscript presented in an intelligible fashion and written in standard English?

Reviewer #1: (No Response)

6. Review Comments to the Author

Reviewer #1: (No Response)

7. PLOS authors have the option to publish the peer review history of their article (what does this mean?). If published, this will include your full peer review and any attached files.

Reviewer #1: **Yes: **Jorge Antonio Coronado Daza

---

## [Editor Report · Acceptance letter]

4 Aug 2022

PONE-D-22-07326R1 

Quality and variation of care for chronic kidney disease in Swiss general practice: A retrospective database study 

Dear Dr. Jäger:

I'm pleased to inform you that your manuscript has been deemed suitable for publication in PLOS ONE. Congratulations! Your manuscript is now with our production department. 

Kind regards, 

on behalf of

Associate Professor Vipa Thanachartwet 

Academic Editor

PLOS ONE